# Mixed Biopolymer Systems Based on Bovine and Caprine Caseins, Yeast β-Glucan, and Maltodextrin for Microencapsulating Lutein Dispersed in Emulsified Lipid Carriers

**DOI:** 10.3390/polym14132600

**Published:** 2022-06-27

**Authors:** Adela Mora-Gutierrez, Sixto A. Marquez, Rahmat Attaie, Maryuri T. Núñez de González, Yoonsung Jung, Selamawit Woldesenbet, Mahta Moussavi

**Affiliations:** 1Cooperative Agricultural Research Center, Prairie View A&M University, Prairie View, TX 77446, USA; rattaie@pvamu.edu (R.A.); mtnunez@pvamu.edu (M.T.N.d.G.); yojung@pvamu.edu (Y.J.); sewoldesenbet@pvamu.edu (S.W.); mamoussavi@pvamu.edu (M.M.); 2Department of Horticultural Sciences, Texas A&M University, College Station, TX 77843, USA; sixto46@tamu.edu

**Keywords:** biopolymers, microencapsulation, spray drying, lutein, emulsified lipid carriers, storage stability

## Abstract

Lutein is an important antioxidant that quenches free radicals. The stability of lutein and hence compatibility for food fortification is a big challenge to the food industry. Encapsulation can be designed to protect lutein from the adverse environment (air, heat, light, pH). In this study, we determined the impact of mixed biopolymer systems based on bovine and caprine caseins, yeast β-glucan, and maltodextrin as wall systems for microencapsulating lutein dispersed in emulsified lipid carriers by spray drying. The performance of these wall systems at oil/water interfaces is a key factor affecting the encapsulation of lutein. The highest encapsulation efficiency (97.7%) was achieved from the lutein microcapsules prepared with the mixed biopolymer system of caprine α_s1_-II casein, yeast β-glucan, and maltodextrin. Casein type and storage time affected the stability of lutein. The stability of lutein was the highest (64.57%) in lutein microcapsules prepared with the mixed biopolymer system of caprine α_s1_-II casein, yeast β-glucan, and maltodextrin, whereas lutein microcapsules prepared with the biopolymer system of bovine casein, yeast β-glucan, and maltodextrin had the lowest (56.01%). The stability of lutein in the lutein microcapsules dramatically decreased during storage time. The antioxidant activity of lutein in the lutein microcapsules was closely associated with the lutein concentration.

## 1. Introduction

The eye health market is expanding in the world. With rising screen times and oxidative stress on eyes of all age groups, the market has moved beyond its traditional focus on seniors to meet the demands of younger people. The changing demographics have prompted food companies to explore new delivery methods for lutein to address the needs of younger consumers. The COVID-19 pandemic strengthened the forces behind the expansion of the eye health market. As more meetings went virtual and working from home became the norm, people began spending more time in front of computers and smartphone screens. Indeed, the COVID-19 pandemic will cause a lasting change in working habits, with the remote model persisting to an extent even after the crisis ends. Changes in screen time have implications for eye health. There is evidence that the high-energy blue light emitted by visual screen devices can cause eye strain and dry eyes [1]. Over the longer term, researchers have linked the cumulative effect of blue light to an increase in the risk of age-related macular degeneration [2,3]. 

The various applications of lutein, a xanthophyll carotenoid with strong antioxidant activity in functional foods, nutraceuticals, and pharmaceuticals are limited due to its insolubility in aqueous media and its little absorption after oral administration and poor bioavailability [4]. Numerous delivery vehicles have been investigated to incorporate lutein into functional foods, nutraceuticals, and pharmaceuticals [4]. However, lutein exhibits low stability against adverse environmental conditions [5]. Hence, the encapsulation approach of lutein decreases its sensitivity and serves as an effective delivery system. The protective mechanism of encapsulation is to form a membrane (wall system) around oil droplets or particles of encapsulated material (core). Some of the encapsulation technologies include spray drying, spray chilling-cooling, fluidized-bed coating, and inclusion of complexation [6]. Recently, microfluidic jet spray drying (MFJSD) has been presented as an efficient microencapsulation technology for the protection and the delivery of visually-beneficial compounds such as carotenoids and omega-3s [7]. Although lutein microcapsules prepared by spray drying cannot effectively decrease the chemical degradation of lutein [7,8], microencapsulation with appropriate wall systems can increase the shelf-life of lutein. 

Natural biopolymer-based micellar delivery systems such as milk proteins (casein, whey proteins) and polysaccharides (β-glucan, chitosan, ι-carrageenan) are high-value wall systems for the encapsulation of hydrophobic bioactive compounds [9,10,11,12,13,14]. In this context, the curcumin-casein micelle complexation could serve as a potential carrier of curcumin for cancer therapy [9]. To be effective in treating cancer, curcumin must be present throughout the affected tissue (s) at high concentration for a sustained period so that it may be taken up by the cancer cells, but not at so high a concentration that normal cells are injured beyond repair [15]. A more effective method of administering a cancer therapeutic, particularly curcumin, is in the form of a dispersion of oil in which curcumin is dissolved. Emulsions have typically been the most cost-effective and gentle to administer curcumin [16,17]. However, a delivery system provides a means for encapsulating the hydrophobic phytochemical curcumin in a hydrocolloid matrix such that when the delivery system is incorporated into pharmaceuticals, the release of curcumin is prolonged, controlled, and able to retain anti-cancer activity, a major biological activity of curcumin [18,19]. To maximize multi-drug therapy, carriers for concomitant administration of multiple bioactive agents have been developed. For instance, the generation of 2-hydroxymethyl starch microparticles for the co-delivery and the controlled release of multiple drug agents is needed within the pharmaceutical industry to target chronic diseases such as cancer and treat infections [20]. A biopolymer-based carrier designed for drug delivery must be biocompatible and degraded products must be nontoxic. Among the biopolymers, nanofibrous aerogels based on cellulose nanofibers and chitosan nanofibers have been demonstrated to be both biocompatible and biodegradable [21]. Porous organic polymers derived from nanopalladium catalysts have also been proposed for the chemo-selective synthesis of antitumor drugs [22]. 

The superior interfacial functionality of caprine casein has been proven in emulsions with added lutein [23] and resveratrol [24]. The lipid carriers emulsified with caprine casein that is naturally rich in β-casein influence the structure of the interface, leading to a closed compact interfacial structure ideal for encapsulation of hydrophobic bioactive compounds [25]. The preparation of emulsions with lutein dispersed in lipid carriers in combination with the mixed biopolymer systems of caprine α_s1_-I casein, yeast β-glucan, and maltodextrin; caprine α_s1_-II casein, yeast β-glucan, and maltodextrin; or bovine casein, yeast β-glucan, and maltodextrin and the subsequent operating conditions of spray drying form the basis of the process of microencapsulation carried out in the present study. It is a natural way to encapsulate and deliver lutein. Indeed, using lutein as a model hydrophobic ingredient, the bovine and caprine casein micelles could be useful as vehicles for the entrapment, protection, and delivery of sensitive ingredients. Such microcapsules may be incorporated into food products, without modifying their sensory properties. In combination with β-glucan, a natural biopolymer with immunomodulatory activity [26], colloidally stable complexes of lutein may be produced. The inclusion of maltodextrin, a versatile source of carbohydrates, is important in obtaining complete encapsulation of lutein after spray drying [27]. 

Preparation of lutein microcapsules not only improves the stability of lutein but also expands the scope of applications of lutein in the food industry. The focus of this study will be the relationship between the stability of lutein and the interfacial properties of the created lutein-loaded emulsion droplets for the preparation of lutein microcapsules. Therefore, the objectives of this study were to examine the lutein-loaded emulsion properties and the properties of spray-dried microencapsulated lutein powders prepared with the mixed biopolymer systems of caseins (bovine, caprine), yeast β-glucan, and maltodextrin as wall systems.

## 2. Materials and Methods

### 2.1. Materials

A commercial preparation of lutein consisting of 20% lutein (*w*/*w*) dissolved in corn oil was a kind gift of Farbes Brands (Park Ridge, NJ, USA). The lutein analytical standard was purchased from Sigma-Aldrich (St. Louis, MO, USA). Medium-chain triglycerides (MCTs) oil (Neobee^®^ M-5, ≥66% C8:0 and ≥32% C10:0 content) was kindly donated by Stepan Company (Northfield, IL, USA). A commercial preparation of yeast β-glucan consisting of a minimum of 30% β-glucan (β-1,3/1,6-glucans) from *Saccharomyces cerevisiae* was a kind gift of Lallemand Bio-Ingredients (Montreal, Canada). Maltodextrin (DE = 18) was obtained from Grain Processing Corporation (Muscatine, IA, USA). The solvents used for extraction and analysis of samples such as hexane, acetonitrile, methanol, and dichloromethane were HPLC-grade and purchased from Sigma-Aldrich. All other chemicals and reagents were of analytical grade and purchased from Sigma-Aldrich. Deionized water, prepared by passing distilled water over a mixed bed cation-anion exchanger, was used throughout this study.

### 2.2. Preparation of Bovine and Caprine Caseins

The caprine caseins characterized by a high content of α_s1_-casein (type I) and α_s1_-casein (type II) were obtained from a French-Alpine goat and an Anglo-Nubian goat, respectively [28] from the International Goat Research Center (IGRC) at Prairie View A&M University. Bovine milk was obtained from a Jersey cow from a nearby farm. Briefly, casein was prepared from skimmed milk by precipitating at pH 4.6 at 30 °C, then neutralized to pH 7.0, dialyzed against deionized water, and lyophilized. The integrity of the samples was confirmed by sodium-dodecyl sulfate-polyacrylamide gel electrophoresis (SDS-PAGE), and densitometry was used to assess the relative concentration of casein components [29].

### 2.3. Emulsion Preparation

The following combinations of mixed biopolymer systems were investigated: bovine casein (50 g/kg), yeast β-glucan (50 g/kg), and maltodextrin (575 g/kg); caprine α_s1_-I casein (50 g/kg), yeast β-glucan (50 g/kg), and maltodextrin (575 g/kg); and caprine α_s1_-II casein (50 g/kg), yeast β-glucan (50 g/kg), and maltodextrin (575 g/kg). Each biopolymer system was dispersed in 1 L of deionized water and then incubated in a water bath at 50 °C, with constant shaking for 20 min. After the mixtures were completely dissolved, the aqueous solutions were left overnight at 4 °C to ensure complete hydration. Lutein from Marigold flower (*Tagetes erecta*) extract (20%, *w*/*w*) dispersed in corn oil (25 g/kg of powder) was further dispersed in MCTs oil (325 g/kg of powder), including oil-phase emulsifiers [di- and mono-glycerides (18 g/kg of powder) and sodium stearoyl lactate (7 g/kg of powder)], were mixed with the biopolymer-based solutions and then pre-emulsified for 3 min using a hand-held homogenizer (Biospec Products Inc., Bartlesville, OK, USA). Coarse emulsions were further homogenized five times at 82.74 MPa (12,000 psi) and at a temperature of 50 °C through a high-pressure TC5 homogenizer (Stansted Fluid Power, Harlow, UK). Before spray drying, the emulsions were cooled down to room temperature.

### 2.4. Viscosity of the Emulsions

Viscosity measurements of the lutein-loaded emulsions were carried out in triplicate with a Mojonnier-Doolittle (Mojonnier Bros, Co., Chicago, IL, USA) viscosimeter at room temperature.

### 2.5. Droplet Size of the Emulsions

The average droplet size of the lutein-loaded emulsions was measured at 0 days of storage (at 20 °C) with a SALD-2101 laser diffraction particle analyzer (Shimadzu, Columbia, MD, USA), after homogenization. A small volume (100 μL) of the lutein-loaded emulsions was diluted in 50-mL of deionized water to obtain a good signal in the detector.

### 2.6. Encapsulation of the Emulsions by Spray Drying

The homogenized lutein-loaded emulsions were spray-dried in a laboratory scale spray drier (Armfield LTD. Hampshire, UK) equipped with a 1.5 mm diameter nozzle atomizer. The spray dryer has an evaporation rate of 1500 and 2000 mL/h. The inlet and the outlet temperatures were maintained at 160 and 90 °C, respectively. The lutein-loaded emulsions were fed into the main chamber through a peristaltic pump with a feed-flow of 7.5 mL/min and a compressor air pressure of 0.15 MPa. The obtained spray-dried powders were collected from the collecting chamber, closed hermetically, and stored at −21 °C before further examination.

### 2.7. Encapsulation Efficiency

Encapsulation efficiency (EE) of the dried powders was evaluated according to the method of Carneiro et al. [30]. A total of 0.5 g of powder was transferred into a 50-mL centrifuge tube, followed by the addition of 5 mL of hexane. The mixture was gently mixed (vortexed) for 2 min to extract the surface oil and then filtered with a Whatman #1 filter paper (Whatman International Ltd., Maidstone, UK), and the collected powder on the filter was rinsed three times with 20 mL of hexane. The filtered powder was rinsed three times, each with 5 mL of hexane. All the hexane was combined and removed from the extracted oil by nitrogen-flow.

The encapsulation efficiency (EE) of the emulsion droplets was calculated as follows:EE = [(Total oil − Surface oil/Total oil)] × 100%(1)

### 2.8. Scanning Electron Microscopy (SEM) Analysis of the Microcapsules

The microstructures of the lutein microcapsules prepared with the mixed biopolymer systems were evaluated at 0 weeks of storage (at 20 °C and 32% relative humidity) by scanning electron microscopy (SEM) using a JEOL JSM-6010LA In TouchScope™ multiple touch panel scanning electron microscope (JEOL Ltd., Tokyo, Japan). Powder samples were mounted on brass stub using double-sided tape and coated with a thin film palladium-gold layer. The backscattered electron images were collected using an accelerating voltage of 7 kV and a load current of ~90 μÅ, with a working distance of 9 mm, at a variable pressure of 10 Pa. The micrographs were collected at a magnification of 1000×.

### 2.9. Structural Characterization of Lutein in the Microcapsules

Fourier Transform Infrared (FTIR) spectra of the lutein extracted from lutein microcapsules at 0 weeks of storage and the reference standard were recorded in the range between 400 and 4000 cm^−1^ with 35 scans per sample, using a FTIR spectrometer (Nicolet Model 740, Madison, WI, USA) equipped with a Nicolet 660 data system. For FTIR measurements, the spray-dried powders containing lutein (200 mg) were mixed with acetone until complete sample discoloration. The samples were filtered through a phase separating filter paper (Whatman IPS), and the resulting extract was concentrated under vacuum (at 6.7 kPa) at room temperature in a rotary evaporator (Büchi Rotavapor^®^ R-210, Büchi Co, New Castle, DE, USA) until a final volume of 5 mL. The acetone was removed from the extracted lutein by nitrogen-flow. The FTIR measurements were carried out under an infrared (IR) lamp.

### 2.10. Moisture Content of the Microcapsules

Two grams of the lutein microcapsules were weighed and dried by oven drying at 70 °C for 24 h to constant weight. The moisture content was measured by the difference in weight before and after drying.

### 2.11. Storage Conditions of the Microcapsules

The storage stability of lutein microcapsules was evaluated at 6 weeks of storage at 20 and 40 °C. Powder samples were spread on glass Petri dishes and kept open to air exposure. All samples were placed in a desiccator with a saturated MgCl_2_-6 H_2_O solution to provide a relative humidity of approximately 32% at 20 and 40 °C in an oven in the dark. The stability of lutein was monitored regularly, and experiments were carried out in triplicate.

### 2.12. HPLC Analysis of Lutein in the Microcapsules

Lutein was analyzed and quantified by high-performance liquid chromatography (HPLC), according to the method of Gong et al. [31], with minor modifications. To measure lutein, 0.1 g of the lutein microcapsules were added to 3 mL of methanol in a capped 4 mL-tube and mixed thoroughly using a vortex mixer. The supernatant was filtered (Whatman #541), collecting approximately 2 mL of filtrate. An aliquot (1 mL) of the filtrate was collected and passed through a 0.45 μm filter using a syringe (Hamilton Company, Reno, NV, USA) into a 2 mL amber vial. The sample solution (10 μL) was injected into the HPLC system. The HPLC (Waters Corporation, Milford, MA, USA) instrument was equipped with a 515 pump, a Waters auto-sampler model 717, a Waters C-30 column (150 mm × 4.6 mm, I.D., 5 μm), and a UV-visible photodiode array detector, model 2489. The Empower 2 software program was used for analysis of the collected data. The isocratic mobile phase consisted of methanol/methylene chloride/acetonitrile (50/20/30). The flow rate was 1 mL/min, the column temperature was at 27 °C for 12 min, and the wavelength was 450 nm. The calibration curve of lutein was constructed using the lutein analytical standard at different dilutions (ranging from 20 μg to 1000 μg/mL), with a correlation coefficient of 0.999. All samples and standards were protected from light to avoid photo-isomerization and oxidation of the lutein.

### 2.13. Antioxidant Activity of Lutein in the Microcapsules

For antioxidant activity measurements, the spray-dried powders containing lutein (200 mg) were mixed with acetone until complete sample discoloration. The samples were filtered through a phase separating filter paper (Whatman IPS), and the resulting extract was concentrated under a vacuum (at 6.7 kPa) at room temperature in a rotary evaporator (Büchi) until a final volume of 5 mL. These extracts were thoroughly mixed with methanol and dilutions (ranging from 1000 to 8000 μg/mL of lutein for each mixed biopolymer system with corresponding dilutions).

A blue-green ABTS radical cation chromophore (ABTS**^+^**) was prepared according to Re et al. [32]. The ABTS**^+^** solution was diluted in 50 mL/100 mL ethanol in water to 0.70 ± absorbance at 734 nm using a Beckman UV/vis model DU-530 spectrophotometer (Beckman Instruments Inc., Fullerton, CA, USA). A diluted ABTS**^+^** solution (3 mL) was added to 30 μL of each treatment or a Trolox standard (final concentration 100–2000 μmol/L in ethanol suspensions); then, an absorbance reading was taken 6 min after the initial mixing at 734 nm (Beckman). The samples’ absorbance measurements were plotted as a function of their concentrations, and they were used to calculate the Trolox Equivalent Antioxidant Capacity (TEAC), where the absorbance reading was used to find the sample concentration equivalent to 1000 μmol/L of Trolox. Results were expressed in μmol/L of Trolox equivalent/g lutein.

### 2.14. Evaluation of Lipid Oxidation in the Microcapsules

Oxidative stability of the lutein microcapsules was examined after storage at 40 °C and 32% relative humidity for 6 weeks. Oxidation was measured using the peroxide value (PV) method and the thiobarbituric acid (TBA) assay. The peroxide values of the lutein microcapsules were determined as follows: a total of 0.5 g of sample powder was mixed with 5 mL of water for 30 min. An aliquot (600 μL) of the aqueous solution was vortexed with a mixture of 3 mL hexane and isopropanol (2:1) for 10 s. The suspension was centrifuged at 1000× *g* for 4 min. An aliquot (200 μL) of the supernatant was treated according to the method of Shantha and Decker [33]. After 20 min, the absorbance was measured at 510 nm (Beckman). The concentration of hydroperoxides was calculated from a hydroperoxide standard curve.

A TBA assay was carried out according to the method of Di Giorgio et al. [34]. A small sample of the lutein microcapsules (0.5 g) was shaken with 1.78 mL of 5% *w*/*v* trichloroacetic acid (TCA) for 30 min, followed by centrifugation (10,000× *g*, 10 min) to obtain a clear supernatant. An aliquot (0.5 mL) of the supernatant was transferred to a 10-mL glass tube and mixed with 0.5 mL of 0.5% *w*/*v* TBA solution that was freshly made. The mixture was incubated with shaking at 70 °C for 30 min and then immediately immersed in an ice-water bath for 15 min. After reaching room temperature, absorbance was measured at 523 nm (Beckman). The concentration of malondialdehyde (MDA) was calculated from an MDA standard curve.

### 2.15. Determination of Caprylic Acid and Capric Acid Fatty Acids in the Microcapsules

To measure caprylic acid (C8:0) and the capric acid (C10:0) content of the MCTs lipid carrier oil present in the lutein microcapsules, a small amount (0.1 g) of the powder samples was added to 875 μL of hexane in a capped 4 mL-tube and then mixed thoroughly using a vortex mixer. The organic phase was separated, methylated, and then analyzed for its fatty acid composition by gas chromatography/mass spectrometry (GC/MS) (Agilent 7890A gas chromatograph, Santa Clara, CA, USA) as previously described (Armah-Agyeman et al. [35]).

### 2.16. Statistical Analysis

In each experiment, the results of triplicate analyses were used to test the mean experimental variables. The data are expressed as the least-square means and standards error. The persistence of lutein and the antioxidant activity of lutein data were analyzed on the two-factorial design, 3 (casein type) × 3 (storage time) using PROC MIXED of SAS (version 9.4., SAS Institute, Cary, NC, USA). Tukey multiple comparison test was applied to find significant differences (*p* < 0.05) among the main effects and their interactions of treatment and storage time.

## 3. Results and Discussion

### 3.1. Characterization of Bovine and Caprine Caseins

The main components of the bovine and the caprine caseins used in our study, i.e., α_s1_-casein, β-casein are listed in Table 1. Caprine caseins in contrast to bovine caseins vary considerably in the types of caseins present: some are poorer in α_s1_-casein and some are richer in α_s1_-casein [28]. The high content of β-casein in caprine caseins (Table 1) makes them suitable candidates as biopolymer matrixes for lipophilic molecules such as *n*-3 fatty acids [36]. The specificity of β-casein for such interactions has been ascribed to its high hydrophobicity, which was evidenced by ^31^P-NMR [25]. It has been previously reported that caprine α_s1_-I casein and caprine α_s1_-II casein appear to bind to lutein, thereby improving stability of lutein [23]. Such differences in β-casein content exhibited by the caprine α_s1_-I casein and caprine α_s1_-II casein were also reflected in the improved stability of resveratrol in oil-in-water emulsions [24]. 

### 3.2. Physical Properties of Emulsions

Droplet size is an important parameter for determining the physical stability of the prepared lutein-loaded emulsions. Table 2 presents the average droplet size of the lutein-loaded emulsions prepared with the biopolymer system of casein (bovine, caprine), yeast β-glucan, and maltodextrin. These results clearly indicate the excellent emulsifying properties of these milk proteins, which include reducing the interfacial tension and the rapid absorption at the oil–water interface during emulsion preparation [23,24]. Low interfacial tension at the oil–water interface is of paramount importance to obtain dispersions with small average droplet size and hence good physical stability. The droplet sizes of the three lutein-loaded emulsions were not significantly different (*p* > 0.05). Likewise, no differences in viscosity were found among the three lutein-loaded emulsions (*p* > 0.05). The low viscosities observed in the three lutein-loaded emulsions facilitate their atomization and spray drying into powders. 

### 3.3. Physical Properties of Microcapsules

Moisture content relates to drying efficiency and powder flowability. Most importantly, moisture content contributes to the destabilization of biological systems during long-term storage [37]. Although the dried lutein microcapsules contain some moisture because of the hydrophilic nature of the biopolymer systems (Table 2), appropriate storage of these lutein microcapsules will ensure flowability of the lutein microcapsules, while prolonging the shelf-life of lutein. 

Encapsulation efficiency (EE) is determined by factors such as the ratio of the core [e.g., the lipophilic phytochemical lutein dispersed in emulsified lipid carriers to a wall system (e.g., the mixed biopolymer system of casein (bovine, caprine), β-glucan, and maltodextrin)]; feed solids concentration; inlet air temperature; and drying air flow. The EE of the lutein microcapsules was more than 90% for all samples prepared from the three types of mixed biopolymer systems (Table 2). These results indicate that optimum operating conditions were achieved to encapsulate the lipophilic phytochemical lutein dispersed in emulsified lipid carriers. The high values of EE obtained in our study may be related to the properties of the mixed biopolymers used as wall systems. Thus, lutein microcapsules prepared with the mixed biopolymer system of caprine α_s1_-II casein, β-glucan, and maltodextrin attained the highest EE (*p* < 0.05). This EE increase may be attributed to the different casein compositions (Table 1). This fact allows us to postulate that β-casein is capable of acting as a stabilizer in oil-in-water emulsions prepared with the caprine caseins, particularly the caprine α_s1_-II casein (Table 1). The β-casein molecule is absorbed on the surface of oil droplets, which is induced by protein unfolding, thereby changing the protein structure and causing a stable and a thick layer over the oil droplets [25,38]. The protein moiety of β-glucan helps in embedding the carbohydrate moiety of this polysaccharide onto the oil-water interface, thereby facilitating emulsification of lipids as previously observed in emulsions stabilized by soy soluble polysaccharides [39,40]. The carbohydrate, due to its high number of hydrogen and hydroxyl groups and their electrostatic interactions, is all external to the protein core, and it accounts for a high concentration of sugar residues at the protein surface.

### 3.4. Morphology of Microcapsules

The morphology of microcapsules determines stability, functionality, and flowability of the powder samples. The occurrence of dents, cracks, and pores on the surface of the microcapsules has a detrimental effect on functionality, reconstitution properties, and the shelf-life provided by the carrier matrix. Figure 1a displays the scanning electron micrograph (SEM) image for the sample from lutein microcapsules prepared with the mixed biopolymer system of bovine casein, yeast β-glucan, and maltodextrin. Similar microstructures by SEM were observed for the samples from the lutein microcapsules prepared with the mixed biopolymer system of caprine α_s1_-I casein, yeast β-glucan, and maltodextrin and the mixed biopolymer system of caprine α_s1_-II casein, yeast β-glucan, and maltodextrin (Figure 1b,c, respectively). All lutein microcapsules prepared with the mixed biopolymer systems were spherical or nearly spherical, which reveal that the lutein microcapsules were prepared by spray drying [41]. The lutein microcapsules prepared with the mixed biopolymer systems also exhibited a smoother surface, suggesting a better retention of the core inside the carrier matrix. In our study, the mixed biopolymer systems based on bovine casein, caprine α_s1_-I casein, or caprine α_s1_-II casein, yeast β-glucan, and maltodextrin may minimize surface cracks of the lutein microcapsules. The absence of cracks is critically important to wall functionality in limiting the oxidative degradation of the lipophilic phytochemical lutein dispersed in emulsified lipid carriers during storage. In this study, the bovine and caprine caseins function as emulsifiers, and they have film-forming properties that contribute to producing lutein microcapsules with fewer surface dents in combination with maltodextrin that forms lutein-loaded emulsions with low viscosity at a high solid content. The health-beneficial yeast β-glucan [26,42] acts as a potential biopolymer matrix in clean label products to replace Arabic gum.

### 3.5. Structural Characterization of Lutein in the Microcapsules

The stability of the lutein extract in the lutein microcapsules was confirmed by FTIR. The FTIR spectra of the lutein extracted from the lutein microcapsules and the lutein reference standard are presented in Figure 2. They give information on the molecular structures of the lutein extract in the lutein microcapsules and the lutein referent standard. The characteristic absorption bands of the lutein referent standard appeared at 2924 cm^−1^ and 2852 cm^−1^, denoting asymmetric and symmetric stretching vibrations of CH_2_ and CH_3_. The absorption band at 1718 cm^−1^ is mainly C=O stretching vibrations. The adsorption band at 3432 cm^−1^ is assigned to intermolecular hydrogen bonding. The frequencies are closer to those obtained with the lutein extracted from the lutein microcapsules, suggesting analogies in the functional group environment. Therefore, the FTIR data validate the molecular structure of the lutein extract in the lutein microcapsules at 0 weeks of storage.

### 3.6. Stability of Lutein in the Microcapsules

The stability of lutein refers to how readily the lutein molecule can undergo chemical reactions that modify the unsaturated polyenic hydrocarbons [43]. Lutein is especially sensitive to light and to thermal stress, particularly in the presence of oxygen. The HPLC results indicated that oxidative degradation of lutein occurred in all lutein microcapsules due to the high sensitivity of its chemical structure when exposed to heat, oxygen, and changes in the moisture content during the processing steps and the storage at 20 °C and 32% relative humidity (Table 3). The stability of lutein was strongly influenced by the mixed biopolymer system used. Thus, the mixed biopolymer system of caprine α_s1_-II casein, yeast β-glucan, and maltodextrin appears to be more effective as a wall system than the mixed biopolymer system of bovine casein, yeast β-glucan, and maltodextrin, and it is the most suitable for stabilizing microcapsules containing the lipophilic phytochemical lutein dispersed in emulsified lipid carriers compared to other mixed biopolymer systems (Table 3). Although lutein-encapsulated formulations were manufactured to safeguard lutein from degradation, the results indicate that there is a dramatic reduction in the content of lutein in the lutein microcapsules during the study period (Table 3). It should be noted here that the high proportion of the emulsified lipid carrier medium-chain triglycerides (MCTs) oil (325 g/kg of powder) *vs*. the low proportion of the emulsified lipid carrier corn oil (20 g/kg of powder) could also be correlated with the stability of lutein (5 g/kg of powder) in the lutein microcapsules (Table 3). The fatty acids of the emulsified lipid carrier corn oil being the least oxidative stable whereas the fatty acids of the emulsified lipid carrier MCTs oil could be considered as the most oxidative stable.

Based on the retention time and intensity of the lutein analytical standard, the persistence of the lutein extract in the lutein microcapsules was quantified at 0 weeks, 3 weeks, and 6 weeks of storage at 20 °C and 32% relative humidity. Significant differences (*p* < 0.05) were observed between the lutein microcapsules at 0 weeks of storage and lutein microcapsules at 3 weeks and at 6 weeks of storage at 20 °C and 32% relative humidity. Lutein was significantly (*p <* 0.05) less stable in lutein microcapsules prepared with the mixed biopolymer system of bovine casein, yeast β-glucan, and maltodextrin (Table 3). Caprine caseins, which show a higher hydrophobic character due to their high content of β-casein (Table 1), interact strongly with lutein [23]. Therefore, the binding of the lipophilic phytochemical lutein to the caprine casein (α_s1_-I) and caprine casein (α_s1_-II) is likely to be driven by an interaction with caprine β-casein [25]. Although yeast β-glucan may form colloidal assemblies with lutein, research is needed to elucidate its potential role in the stabilization of lutein.

### 3.7. Antioxidant Activity of Lutein in the Microcapsules

The antioxidant activity of the lutein extract in the lutein microcapsules was assessed by the ABTS free radical method [32], which shows a reduction in antioxidant activity upon increasing the storage time from 0 weeks to 6 weeks at 20 °C and 32% relative humidity (Table 4). The structural changes to the isoprene backbone of lutein have a detrimental effect on the electron movement among the conjugated double bonds [43], thereby lowering the ability of lutein to function as an antioxidant. The ABTS radical reacts with carotenoids with increasing reactivity based on their chemical structure (i.e., number of conjugated double bonds, ionization potential, etc.) [44]. A high Trolox equivalent number indicates a high antioxidant activity as compared to a standard antioxidant (Trolox) [32]. In this study, the antioxidant measurement results are directly proportional to the concentration of the lutein extract in the lutein microcapsules (Table 3 and Table 4). 

Additionally, the antioxidant activity of the lutein extract in the microcapsules assessed by the ABTS method showed a significant (*p* < 0.05) high antioxidant activity in lutein microcapsules prepared with the mixed biopolymer system of caprine α_s1_-I casein, yeast β-glucan, and maltodextrin and the mixed biopolymer system of caprine α_s1_-II casein, yeast β-glucan, and maltodextrin compared with the mixed biopolymer system of bovine casein, yeast β-glucan, and maltodextrin (Table 4). Free lutein exhibited an ABTS value of 705.3 μmol/L TE/g (data not shown), confirming that the observed results are likely due to the lutein extract encapsulated by the three mixed biopolymer systems (Table 4). Before spray drying, lutein-loaded emulsions emulsified by bovine casein or caprine casein were absorbed into oil/water interfaces. The unique compact structure of the lutein-loaded emulsions emulsified by the caprine caseins, and the relatively thick surface layer around the oil droplets reduced the exposure of lutein and inhibited its degradation [23], which further retained the antioxidant activity of lutein (this study). 

### 3.8. Accelerated Shelf-Life Tests in the Microcapsules

To determine the shelf-life of the lutein microencapsulates, an accelerated test was used. The lutein microcapsules, which are prepared with MCTs oil at 32.5% oil load/corn oil at 2% oil load, were maintained at 40 °C to accelerate the deterioration reactions. We measured PV and TBA values to predict the outcomes of the oxidation reactions of the lutein microcapsules. Testing was performed at 0 and 6 weeks of storage and the results are presented in Table 5. After 6 weeks of storage at 40 °C and 32% relative humidity, the PV and TBA values of the lutein microcapsules prepared with the biopolymer systems of bovine casein, yeast β-glucan, and maltodextrin; caprine α_s1_-I casein, yeast β-glucan, and maltodextrin; and caprine α_s1_-II casein, yeast β-glucan, and maltodextrin were not statistically different (*p* > 0.05).

Microencapsulation of lutein dispersed in emulsified lipid carriers with the biopolymer systems of bovine casein, yeast β-glucan, and maltodextrin, caprine α_s1_-I casein, yeast β-glucan, and maltodextrin, and caprine α_s1_-II casein, yeast β-glucan, and maltodextrin successfully inhibited oxidation by limiting the PV and TBA values of the emulsified lipid carriers where lutein is dispersed within 3.0 meq/kg oil and 1.0 mg MDA/Kg oil (Table 5). A previous study by Li and Shi [45] clearly indicated that the mixed biopolymer system of bovine casein (50 g/kg), oat β-glucan (50 g/kg), and maltodextrin (575 g/kg) along with borage oil (350 g/kg) and emulsifiers (25 g/kg) shows a high efficiency in controlling the formation of hydroperoxides in borage oil powders subjected to an accelerated storage test for 30 days at 45 °C and 33% relative humidity. The major potential advantage of proteins (e.g., casein, whey protein isolate, soy protein isolate, fish gelatin) as emulsifiers in foods is their ability to protect polyunsaturated lipids from oxidation [46,47]. Moreover, the combination of gelatin and sodium caseinate isolated from bovine milk has been shown to reduce the oxidative degradation of the xanthophyll carotenoid astaxanthin in nanodispersions [48]. Maltodextrin may confer protection against oxidation [49]. 

The amount of caprylic acid (C8:0) and capric acid (C10:0) fatty acids in the lutein microcapsules was measured at 6 weeks of storage (at 40 °C and 32% relative humidity) and ranged from 217-65 μg/μL (Table 6), but GC/MS analysis of the C8:0 and C10:0 fatty acids from the lutein microcapsules at 0 weeks of storage (data not presented) did not reveal any differences in fatty acid content with the fatty acid content obtained at 6 weeks of storage (Table 6). These GC/MS results suggest that lutein microcapsules prepared with MCT oil/corn oil are oxidative stable at 40 °C and 32% relative humidity. Moreover, numerically higher levels of C8:0 and C10:0 fatty acids distinguished the lutein microcapsules prepared with the biopolymer system of caprine α_s1_-II casein, yeast β-glucan, and maltodextrin from the other biopolymer systems used in this study at 6 weeks of storage (Table 6). 

After 6 weeks of storage at 40 °C and 32% relative humidity, all lutein microcapsules showed a low persistence of lutein (Table 7). As mentioned earlier, lutein is prone to oxidation. When lutein is dispersed in edible oils high in polyunsaturated fatty acids (PUFAs), it gets easily oxidized and protects against the oxidation of fatty acids [50]. Likely, lutein dispersed in emulsified lipid carriers, i.e., MCTs oil at 32.5% oil load and corn oil at 2% oil load was oxidized, and it protected against the oxidation of fatty acids, specifically those PUFAs of corn oil. Results show that the mixed biopolymer system of caprine α_s1_-I casein, yeast β-glucan, and maltodextrin, and the mixed biopolymer of caprine α_s1_-II casein, yeast β-glucan, and maltodextrin were more efficient than the mixed biopolymer of bovine casein, yeast β-glucan, and maltodextrin in reducing oxidative degradation of lutein in the lutein microcapsules stored at 40 °C and 32% relative humidity for 6 weeks. The film-forming ability of the bovine and the caprine caseins was not enough to provide stability over time at an accelerated rate (Table 7). However, the mixed biopolymer of caprine α_s1_-II casein, yeast β-glucan, and maltodextrin presented greater protection for lutein, exhibiting a lutein content of 50.79% during the 3-week storage period at 40 °C and 32% relative humidity. Álvarez-Henao et al. [51], who obtained encapsulated lutein using modified food starch (33.3%), Arabic gum (33.3%), and maltodextrin (33.3%) by the spray drying method, had a lutein content of 35% when subjected to an accelerated storage test for 20 days at 40 °C and 75% relative humidity. The combination of modified starch, Arabic gum, and maltodextrin provides a suitable encapsulating matrix for lutein with improved characteristics such as emulsifying, film-forming, and binding properties [51]. As expected, rapid degradation of encapsulated lutein at high storage temperature and high relative humidity was observed. These results demonstrated that lutein encapsulated products in a powder format should be stored in a cool, dry place. 

## 4. Conclusions

Lutein is highly susceptible to oxidative degradation. Hence, improving the stability of lutein is imperative for extending its shelf-life. The HPLC data indicated that microencapsulation with the mixed biopolymer system based on caprine casein (α_s1_-II), yeast β-glucan, and maltodextrin resulted in the highest stability of lutein (64.57%) during 6 weeks of storage at 20 °C and 32% relative humidity. The GC/MS results highlighted a significant impact of the emulsified lipid carriers, i.e., MCTs oil and corn oil on lutein degradation in the lutein microcapsules during 6 weeks of storage at 40 °C and 32% relative humidity. Lutein loss was reduced due to the high oxidative stability of the emulsified lipid carrier MCTs oil, the type of casein used as a wall system (bovine casein, caprine casein), and the high concentration of yeast β-glucan and maltodextrin, which were an effective combination for enhancing the stability of lutein in the lutein microcapsules. Caprine casein is an excellent carrier for hydrophobic bioactive compounds like lutein by forming compact and thick interfacial layers, thereby allowing better encapsulation of lutein. The present study provides a microencapsulation process to produce mixed biopolymer systems based on bovine and caprine caseins, yeast β-glucan, and maltodextrin for the delivery of lutein in foods. 

## Figures and Tables

**Figure 1 polymers-14-02600-f001:**
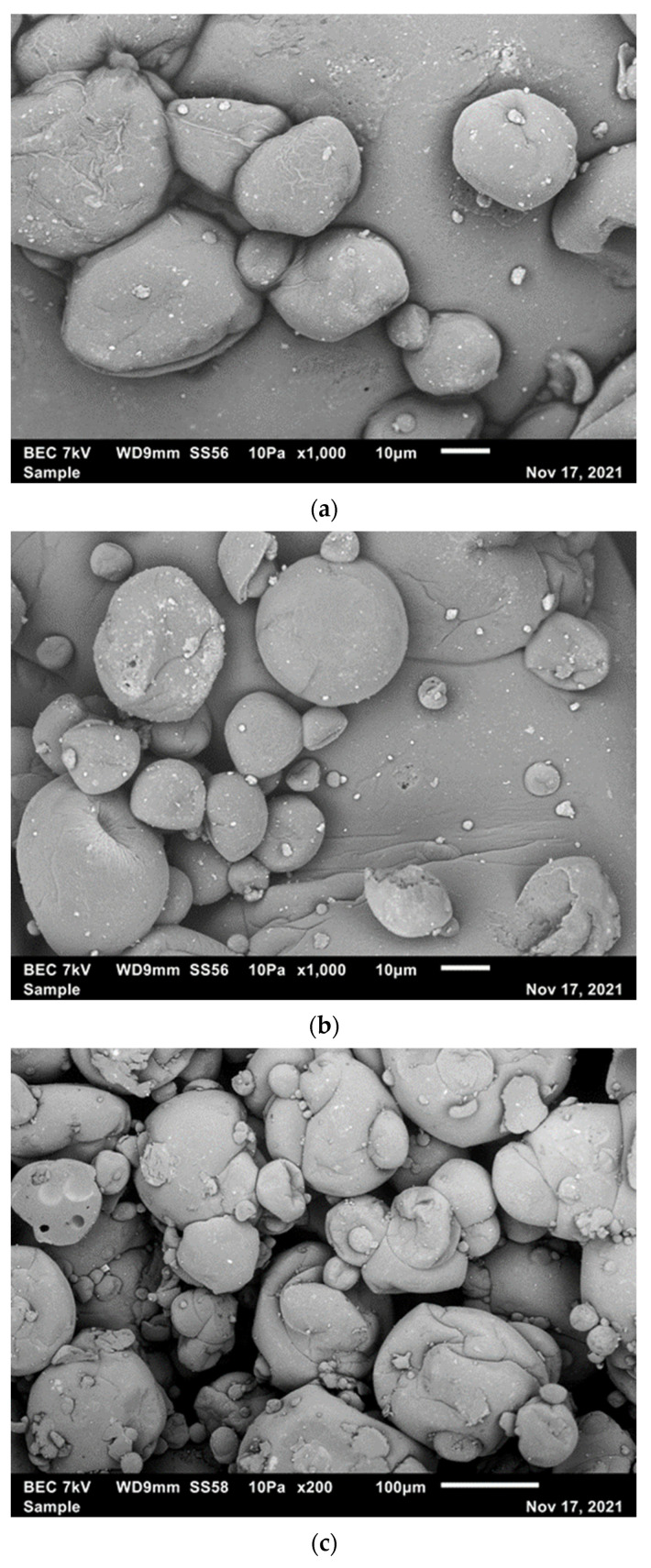
Microcapsules coated with the mixed biopolymer systems of (**a**) bovine casein, yeast β-glucan, and maltodextrin; (**b**) caprine α_s1_-I casein, yeast β-glucan, and maltodextrin; and (**c**) caprine α_s1_-II casein, yeast β-glucan, and maltodextrin at 0 weeks of storage observed with SEM.

**Figure 2 polymers-14-02600-f002:**
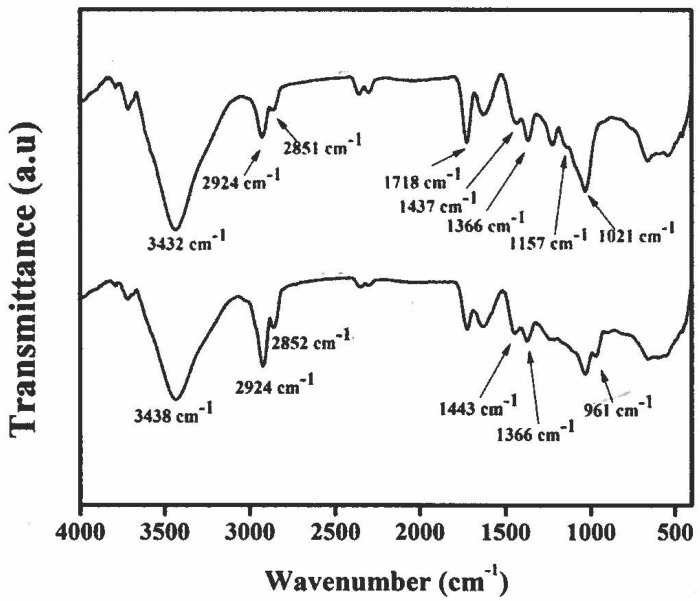
FTIR spectra. (Top) FTIR spectrum of the lutein reference standard. (Bottom) FTIR spectrum of the lutein extract in the microcapsules.

**Table 1 polymers-14-02600-t001:** Casein distribution of caprine caseins compared with a typical bovine casein ^1^.

Casein Fraction, %					
Sample	α_s2_	α_s1_ *	α_s1_ *	β	κ
Caprine casein high in α_s1_-I	9.2	4	21.1	51.6	13.8
Caprine casein high in α_s1_-II	5.3	…	25.6	60.6	9.6
Bovine casein	12.1	…	39.5	37.2	11.2

^1^ Densitometry. * Altered SDS-polyacrylamide gel electrophoresis mobility may represent a truncated version of α_s1_-casein.

**Table 2 polymers-14-02600-t002:** Physicochemical properties of lutein-loaded emulsions and lutein microcapsules.

	Bovine CN + BG + MD ^1^	Caprine (α_s1_-I) CN + BG + MD	Caprine (α_s1_-II) CN + BG + MD
*Emulsion (before drying)*			
Droplet size (nm)	202.00 ± 1.53	197.33 ± 0.67	198.00 ± 1.16
Viscosity (mPa.s)	6.73 ± 0.03	6.82 ± 0.07	6.93 ± 0.05
*Microcapsule (after drying)*			
EE ^2^ (%)	91.07 ± 0.56 ^b^	92.76 ± 1.71 ^b^	97.70 ± 1.35 ^a^
Moisture content (g/Kg)	12.81 ± 0.08 ^a^	11.71 ± 0.22 ^b^	11.52 ± 0.34 ^b^

Least-square mean (LSM) values and standard error (SE). ^a,b^ Means in the same row with different superscripts are different (*p* < 0.05). ^1^ CN, casein; BG, β-glucan; MD, maltodextrin DE = 18. ^2^ EE, encapsulation efficiency.

**Table 3 polymers-14-02600-t003:** Persistence of lutein in the lutein microcapsules during storage for 6 weeks at 20 °C and 32% relative humidity.

Treatment/Storage Time	Persistence of Lutein (%) ^2^
*Microcapsules* ^1^	
Bovine CN-BG-MD	56.01 ± 0.77 ^c^
Caprine (α_s1_-I) CN-BG-MD	60.82 ± 0.77 ^b^
Caprine (α_s1_-II) CN-BG-MD	64.57 ± 0.77 ^a^
*Storage (week)*	
0	91.40 ± 0.77 ^a^
3	69.26 ± 0.77 ^b^
6	20.73 ± 0.77 ^c^

^1^ CN, casein; BG, β-glucan; MD, maltodextrin DE = 18. ^2^ Least-squares means (LSMeans) ± standard error (SE). ^a,b,c^ LSMeans in the same column within each microcapsule and each storage time with different superscripts are different (*p* < 0.05).

**Table 4 polymers-14-02600-t004:** Antioxidant activity of lutein in the lutein microcapsules during storage for 6 weeks at 20 °C and 32% relative humidity by the ABTS free radical method.

Treatment/Storage Time	ABTS (μmol/L TE/g) ^2^
*Microcapsules* ^1^	
Bovine CN-BG-MD	227.81 ± 28.94 ^c^
Caprine (α_s1_-I) CN-BG-MD	327.26 ± 34.22 ^b^
Caprine (α_s1_-II) CN-BG-MD	361.42 ± 38.54 ^a^
*Storage (week)*	
0	425.76 ± 25.47 ^a^
3	298.74 ± 19.00 ^b^
6	191.99 ± 16.25 ^c^

^1^ CN, casein; BG, β-glucan; MD, maltodextrin DE = 18. ^2^ Least-squares means (LSMeans) ± standard error (SE). ^a,b,c^ LSMeans in the same column within each microcapsule and each storage time with different superscripts are different (*p* < 0.05).

**Table 5 polymers-14-02600-t005:** Formation of hydroperoxides and MDA from lutein microcapsules at week 0 and week 6 (at 40 °C and 32% relative humidity).

Microcapsules ^1^	PV (meqO_2_/Kg) ^2^		TBA (mg MDA/Kg) ^2^	
	Week 0	Week 6	Week 0	Week 6
Bovine CN-BG-MD	0.88 ± 0.04 ^a^	2.63 ± 0.23 ^b^	0.19 ± 0.03 ^a^	0.86 ± 0.01 ^b^
Caprine (α_s1_-I) CN-BG-MD	0.87 ± 0.02 ^a^	2.48 ± 0.11 ^b^	0.18 ± 0.01 ^a^	0.81 ± 0.02 ^b^
Caprine (α_s1_-II) CN-BG-MD	0.86 ± 0.02 ^a^	2.35 ± 0.30 ^b^	0.16 ± 0.01 ^a^	0.76 ± 0.04 ^b^

^1^ CN, casein; BG, β-glucan; MD, maltodextrin DE = 18. ^2^ Least-squares means (LSMeans) ± standard error (SE). ^a,b^ Means in the same rows and the same columns for PV and MDA with different superscripts are different (*p* < 0.05).

**Table 6 polymers-14-02600-t006:** Composition (µg/µL) of fatty acid methyl esters from lutein microcapsules at week 6 (at 40 °C and 32% relative humidity).

Microcapsules ^1^	Methyl Octanoate (C8:0) ^2^	Methyl Decanoate (C10:0) ^2^
Bovine CN-BG-MD	217.77 ± 11.93	65.33 ± 2.31
Caprine (α_s1_-I) CN-BG-MD	233.52 ± 13.10	73.50 ± 4.69
Caprine (α_s1_-II) CN-BG-MD	280.96 ± 33.24	80.16 ± 6.86

^1^ CN, casein; BG, β-glucan; MD, maltodextrin DE = 18. ^2^ Mean value ± standard error (SE).

**Table 7 polymers-14-02600-t007:** Persistence of lutein in the lutein microcapsules during storage for 6 weeks at 40 °C and 32% relative humidity.

Treatment/Storage Time	Persistence of Lutein (%) ^2^
*Microcapsules* ^1^	
Bovine CN-BG-MD	42.81 ± 0.427 ^c^
Caprine (α_s1_-I) CN-BG-MD	47.99 ± 0.427 ^b^
Caprine (α_s1_-II) CN-BG-MD	54.26 ± 0.427 ^a^
*Storage (week)*	
0	90.86 ± 0.427 ^a^
3	42.17± 0.427 ^b^
6	12.02 ± 0.427 ^c^
*Interaction of microcapsule type × storage*	
Bovine CN-BG-MD × week 0	86.44 ± 0.740 ^c^
Caprine (α_s1_-I) CN-BG-MD × week 0	91.17 ± 0.740 ^b^
Caprine (α_s1_-II) CN-BG-MD × week 0	94.98 ± 0.740 ^a^
Bovine CN-BG-MD × week 3	34.96 ± 0.740 ^f^
Caprine (α_s1_-I) CN-BG-MD × week 3	40.76 ± 0.740 ^e^
Caprine (α_s1_-II) CN-BG-MD × week 3	50.79 ± 0.740 ^d^
Bovine CN-BG-MD × week 6	7.04 ± 0.740 ^i^
Caprine (α_s1_-I) CN-BG-MD × week 6	12.02 ± 0.740 ^h^
Caprine (α_s1_-II) CN-BG-MD × week 6	17.01 ± 0.740 ^g^

^1^ CN, casein; BG, β-glucan; MD, maltodextrin DE = 18; ^2^ Least-squares means (LSMeans) ± standard error (SE); ^a,b,c,d,e,f,g,h,i^ LSMeans in the same column within each microcapsule, each storage time, and the interaction with different superscripts are different (*p* < 0.05).

## Data Availability

Not applicable.

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
