# Peer review of "Mixed Biopolymer Systems Based on Bovine and Caprine Caseins, Yeast β-Glucan, and Maltodextrin for Microencapsulating Lutein Dispersed in Emulsified Lipid Carriers"

_polymers, 2022, doi:10.3390/polym14132600_

Round 1
Reviewer 1 Report
The paper tried to encapsulate Lutein using mixed biopolymer systems based on casein (bovine, caprine), yeast β-glucan, and maltodextrin by spray-drying. The paper needs to address the following issues:
-please add microencapsulation in keywords
-what is the main result of this study and what did you get? It must be clarified in abstract
-it is suggested to add the following studies to enrich your introduction: https://doi.org/10.1021/acs.orglett.9b01230, https://doi.org/10.1021/acsnano.1c11301, https://doi.org/10.1080/10717544.2021.1955043, https://doi.org/10.1186/s40538-021-00214-x, https://doi.org/10.1016/j.scp.2022.100606
-How did you measure “Surface oil” in the EE% assay?
-why SEM images don’t have a scale bar? Must add
-Figure 2 has low quality of resolution
Author Response
We are grateful for your precious suggestions and comments to our manuscript. They are extremely beneficial to improve the quality of our work. We have studied the comments carefully and revised our manuscript thoroughly. The point-by-point responses are listed below and the majority of them have been incorporated into the revised manuscript.
Comment 1: Please add microencapsulation in key words.
Response: The key word microencapsulation has been added to the revised version of the manuscript.
Comment 2: What is the main result of this study and what did you get? It must be clarified in the Abstract.
Response: Thank you very much for your valuable suggestion. We have added the requested information to the Abstract section (Page 2; Line 34 thru Line 38) in the revised version of the manuscript.
Comment 3: It is suggested to add the following studies to enrich your introduction.
Response: Thank you for providing the link to research studies related to our research study. We have chosen those research studies relevant to our research study. We believe our revised Introduction section will meet your requirement.
Comment 4: How did you measure "surface oil" in the EE % assay?
Response: We have used the method of Carneiro et al. [23] to measure the surface oil. The method disclosed by Carneiro et al. [23] is the method of choice to calculate EE (%). The research study of Carneiro et al. [23] is backed by more than 500 citations. Please refer to sub-section 2.7, Encapsulation Efficiency to know the details of this methodology (Page 7; Line 163 thru Line 165; Page 8; Line 166 thru Line 171) in the revised version of the manuscript.
Comment 5: Why SEM images do not have a scale bar?
Response: Sorry for this technical issue. In the revised manuscript, the SEM images show the scale bar.
Comment 6: Figure 2 has low quality of resolution.
Response: Unfortunately, we could not improve the resolution of Figure 2. Therefore, the HPLC chromatograms (Figure 2) have been deleted in the revised version of the manuscript.
Reviewer 2 Report
The paper submitted by Mora-Gutierez et al. deals with the preparation, by spray-drying starting from emulsions, of a series of lutein-loaded microcapsules based on different combinations of bovine and caprine caseins, yeast glucan and maltodextrin.
The paper is clear, well written but some corrections are needed in order to increase the overall quality:
- the introduction section must be completed with some references concerning the practical applications of the biopolymers used by the authors. A suggestion can be: https://doi.org/10.3390/polym13030477
- the authors state that they have obtain microcapsules but it will be better if the could demonstrate that by using the SEM
- also, the authors state that there are some hydrophobic interactions between lutein and β-casein. In this case, they must carry out some FTIR spectra for free lutein, free microcapsules capsules and lutein-loaded microcapsules. The shift of the characteristic peaks of lutein will proof the presence of these interaction with the polymeric matrix.
- concerning the antioxidant activity, the authors must also add the results for the free lutein. From their results is not clear if the polymeric matrix has a protective role or not! Moreover, the discussion of these results is quite poor. I recommend the comparation with other literature papers concerning the antioxidant activity studies. A suggestion can be: https://doi.org/10.3390/ijms22063075
Author Response
The authors sincerely thank the reviewer for his/her valuable comments and suggestions to improve the quality of the manuscript.
Comment 1: The introduction section must be completed with some references concerning the practical applications of the polymers used by the authors. A suggestion is provided by the reviewer.
Response: The current study was compared with already reported studies of various natural polymers, including the ones used in our study, involved in the preparation of lutein microcapsules (Page 4; Line 77 thru Line 90).
Comment 2: The authors state that they have obtain microcapsules, but it will be better if they could demonstrate that by using the SEM.
Response: The SEM images of the lutein microcapsules are provided on Page 27; Line 604 thru Line 632.
Comment 3: The authors state that there are some hydrophobic interactions between lutein and beta-casein. In this case, they must carry out some FTIR spectra for free lutein, free microcapsules and lutein-loaded microcapsules. The shift of the characteristic peaks of lutein will proof the presence of these interactions with the polymeric matrix.
Response: Thanks a lot for this comment. We do not have free microcapsules and lutein-loaded microcapsules for FTIR measurements. We learned that the yeast beta-glucan required as wall system to prepare microcapsules in the presence and absence of lutein, sold by a Canadian company (Lallemand Bio-Ingredients) and manufactured for this biotech company by a biotech company in Denmark, will take 7-8 weeks to get. In addition, it takes at least 5 weeks to finish the processing of the microcapsules because of the logistic steps involved like requesting access to the spray drier at Texas A&M University. Also, the wall systems of caseins (bovine, caprine type I, caprine type II) need to be manufactured. We have used these dairy ingredients in our previous studies published in J. Dairy Sci. [18] and Foods [19] (References listed in the revised manuscript). Therefore, we do not have these dairy ingredients at this time. The casein composition of cows and goats is influenced by the state of lactation. Thus, the maximum content of individual caseins (Table 1 of the revised manuscript) is observed at early lactation (February thru April). We have to wait until next year to collect the individual milks of cow and goats at early lactation.
Nevertheless, it is useful for us to provide a valuable insight into the interaction of lutein with the wall systems (i.e., lutein interacting with beta-casein in caprine caseins) by making emphasis in our previous research study with lutein-loaded emulsions as inferred by phosphorus-31nuclear magnetic resonance (Mora-Gutierrez et al. [20]. We have stated in the submitted version (and the revised version) of the manuscript that "Although yeast beta-glucan may form colloidal assemblies with lutein, research is needed to elucidate its potential role in the stabilization of lutein" (Page 17; Line 382 thru Line 383).
All in all, we sincerely hope that the interpretation of the interaction taking place between lutein and the caprine caseins, which are high in beta-casein, in the lutein-loaded emulsions for their microencapsulation can be considered and accepted by the reviewer.
Comment 4: Concerning the antioxidant activity, the authors must also add the results of the free lutein. From their results is not clear if the polymeric matrix has a protective role or not! Moreover, the discussion of these results is quite poor. I recommend the comparison with other literature papers concerning the antioxidant activity studies. A suggestion is given by this reviewer.
Response: We thank the reviewer for the comment. We would like to state that this study is new despite the general knowledge about the antioxidant activity of lutein in delivery systems for food and pharmaceutical applications. We are providing lutein along with the weight management ingredient MCTs oil in a powder format.
In the revised version of the manuscript, we are reporting the ABTS method instead of the DPPH method. The ABTS radical reacts with carotenoids with increasing reactivity based on their chemical structure.
It should be noted here that all data presented in the revised version of the manuscript, including the ABTS data, were collected in 2021. The results for the free lutein has been added to the narrative (see sub-section 3.7 in the revised version of the manuscript (Page 18; Line 400 thru Line 402)). This sub-section 3.7 has been worked upon (Page 17 thru Page 18) in the revised version of the manuscript. We thank the reviewer for sharing the article with us. Suitable modifications were made in sub-section 3.7 (Page 17 thru Page 18) with appropriate references.
Reviewer 3 Report
The current manuscript entitled “Mixed Biopolymer Systems Based on Bovine and Caprine Caseins, Yeast β-Glucan, and Maltodextrin for Microencapsulation of Lutein” by Mora-Gutierrez et al demonstrated on the encapsulation of Lutein using mixed biopolymer systems based on casein (bovine, caprine), yeast β-glucan, and maltodextrin by spray-drying. The highest encapsulation efficiency (97.7 %) was achieved from the microcapsules made with the mixed biopolymer system of caprine αs1-II casein, yeast β-glucan, and maltodextrin. The work is good, and it can be useful for the readers of “Polymers”. The manuscript can be accepted for publication after addressing the following comments.
- Provide some more potential information in the introduction section.
- At the end of introduction, mention why you have carried this study and what u have carried out in the present study. Main content of the work is missing.
- Mention the name of organic solvents used in the Materials Section
- Make some discussion on the present work and the existing literature of the related work
- Mention the scale bar for the FESEM images. Scale bar is missing.
- Conclusions is not with adequate information.
Author Response
The authors sincerely thank the reviewer for his/her valuable comments and suggestions to improve the quality of the manuscript.
Comment 1: Provide some more potential information in the Introduction section.
Response: In the revised manuscript, additional information is provided.
Comment 2: At the end of the Introduction section, mention why you have carried this study and what you have carried out in the present study. Main content of the work is missing.
Response: We have re-written and reorganized the introduction section, as you suggested.
Comment 3: Mention the name of organic solvents used in the Material section.
Response: In the revised manuscript, the name of organic solvents used are provided (Page 5; Line 116 thru Line 118).
Comment 4: Make some discussion on the present work and the existing literature of the related work.
Response: We have made big modifications to the revised manuscript from title to research data. We have discussed in detail the present work and related our research findings to existing literature.
Comment 5: Mention the scale bar for the SEM images. Scale bar is missing.
Response: We are sorry for this technical issue. The scale bars have been incorporated to the SEM images.
Comment 6: Conclusions is not with adequate information.
Response: In the revised manuscript, we have improved the Conclusions section.
Round 2
Reviewer 1 Report
Unfortunately, the authors didn't address my comments properly as I asked before. the suggested studies didn't include in the introduction.
Author Response
We thank the reviewer # 1 once again for his/her sincere contribution for the betterment of the manuscript.
Reply to Reviewer # 1:
- We have improved the abstract of the revised version of the manuscript within the 200 words limit.
- We thank the reviewer for sharing the articles with us, but we have included ONLY those articles related to the research presented in the revised version of the manuscript. The current study was compared with already reported studies of various technologies involved in the preparation of lutein-loaded formulations with suitable references [Page 3 thru Page 4; Line 71 thru Line 90]. Please refer to the revised version of the manuscript and ignore the original version of the manuscript.
- We have invited Dr. Harold M. Farrell, Jr, who is a world-leading authority in Dairy Research and collaborates with us in several research projects, to check the manuscript for grammar and syntax as well as clarity of presentation. According to Dr. Farrell, a native English speaker, the revised version of the manuscript is "clear and concise" to the readers.
Reviewer 2 Report
The paper can be published as it is.
Author Response
We acknowledge comments made by Reviewer #2 very much, which were very valuable in improving the quality of our manuscript.